# An Umbrella Review of the Association Between Periodontal Disease and Diabetes Mellitus

**DOI:** 10.3390/healthcare12222311

**Published:** 2024-11-19

**Authors:** Heber Isac Arbildo-Vega, Fredy Hugo Cruzado-Oliva, Edward Demer Infantes-Ruíz, Franz Tito Coronel-Zubiate, Eric Giancarlo Becerra-Atoche, Wilfredo Terrones-Campos, Paul Martín Herrera-Plasencia, Oscar Alex Seminario-Trelles, Roberto Enrique Ortega-Gallegos

**Affiliations:** 1Faculty of Dentistry, Universidad San Martín de Porres, Chiclayo 14012, Peru; hiav30@gmail.com; 2Faculty of Human Medicine, Universidad San Martín de Porres, Chiclayo 14012, Peru; 3Faculty of Stomatology, Universidad Nacional de Trujillo, Trujillo 13001, Peru; fcruzado@unitru.edu.pe; 4Faculty of Health Science, Stomatology School, Universidad César Vallejo, Piura 20001, Peru; einfantesr@ucvvirtual.edu.pe (E.D.I.-R.); ebecerra@ucv.edu.pe (E.G.B.-A.); wterrones@ucv.edu.pe (W.T.-C.); pherrera@ucv.edu.pe (P.M.H.-P.); oseminario@ucv.edu.pe (O.A.S.-T.); reortegag@ucvvirtual.edu.pe (R.E.O.-G.); 5Faculty of Health Sciences, Stomatology School, Universidad Nacional Toribio Rodríguez de Mendoza de Amazonas, Chachapoyas 01001, Peru

**Keywords:** periodontal disease, diabetes mellitus, gestational diabetes, systematic review, meta-analysis

## Abstract

Aim: To determine the clinical association between periodontal disease and diabetes mellitus through an umbrella review. Materials and Methods: A search for publications up to August 2023 was conducted using the following electronic databases: PubMed, Cochrane Database, Scopus, SciELO, Google Scholar, and OpenGrey. We included systematic reviews (SRs) with or without meta-analysis evaluating primary studies that investigated the association between periodontal disease and diabetes mellitus, and there were no time or language restrictions. Literature or narrative reviews, rapid reviews, intervention studies, observational studies, preclinical and basic research, abstracts, comments, case reports, protocols, personal opinions, letters, and posters were excluded. The AMSTAR-2 tool was used to determine the methodological quality of the included studies. Results: The preliminary search yielded a total of 577 articles, of which only 17 remained after discarding those that did not meet the selection criteria. Following their analysis, an association between periodontal disease and diabetes mellitus (type 1 and type 2 diabetes mellitus and gestational diabetes mellitus) was found. Conclusions: The findings and conclusions of this umbrella review indicate with high confidence that periodontal disease is associated with the onset of type 1 and type 2 diabetes mellitus and gestational diabetes.

## 1. Introduction

Diabetes mellitus (DM) is a common metabolic disease resulting from a defect in either insulin secretion or insulin action, or a combination of both [1,2]. This chronic disease is one of the most common in the world, and its prevalence is expected to increase by almost 50% in the coming years. It is estimated that by 2045, there will be around 700 million people living with diabetes, which will be one of the five main causes of death [3]. Mortality in people with diabetes occurs due to complications, of which the most important include microvascular changes, neuropathy, retinopathy, nephropathy, delayed healing, and periodontitis [4,5].

Periodontitis is a disease characterized by chronic inflammation of the entire periodontium that can irreparably destroy the tissue surrounding the tooth, leading to gingival bleeding, increased tooth mobility, and tooth loss [6]. It is estimated that approximately between 20 and 60% of the world’s population suffers from periodontal disease [7,8].

DM and periodontal disease (PD) are associated, despite lacking a common pathophysiology [6]. However, glucose alteration in these patients shows a positive association, which is evident in patients with poorly controlled diabetes having an 86% higher risk of developing periodontitis than non-diabetics or those with well-controlled diabetes [4]. Furthermore, PD may be a risk factor for diabetes mellitus, suggesting a possible bidirectional association [3]. The proposed link between the two diseases is based on characteristics of the immune response, neutrophil function, and cytokine biology [9,10].

The need for dentists to understand the associations and short- and long-term con-sequences of these diseases necessitates a deeper comprehension of the scientific basis using the highest quality evidence available for clinical decision-making [11]. The synthesis of knowledge within a single document can support clinical decision-making, thus facilitating the reading and understanding of a particular topic. To date, no comprehensive synthesis or evaluation of all systematic reviews, including those from recent years, has been performed. Therefore, the aim of this overview is to consolidate the available evidence and address the following question: “What is our current understanding of the link between periodontal disease and diabetes mellitus?” In addition, we assess the general confidence in the systematic reviews analyzing this issue.

## 2. Materials and Methods

### 2.1. Protocol and Registration

The study was executed in accordance with the Preferred Reporting Items for Systematic Reviews and Meta-Analysis Protocols (PRISMA-P) [12] and subsequently registered in the Prospective Registry of Systematic Reviews (PROSPERO) [13]. The registry can be accessed by the general public under the number CRD42023464043. Furthermore, PRIO-harms (Preferred Reporting Items for Overview of Systematic Reviews Checklist) provides the basis for reporting in this study [14]. Notably, our methodology is based on Pauletto et al. [15], given its similarity to our approach. Ethical approval was not required for this umbrella review.

### 2.2. Eligibility Requirements and Noteworthy Results

All included articles were systematic reviews (SRs) that assessed primary research assessing the relationship between PD and DM, regardless of whether meta-analysis was employed and without regard to time or language constraints.

Exclusions included abstracts, remarks, case reports, procedures, personal opinions, letters, posters, fast reviews, intervention studies, observational studies, preclinical and fundamental research, and literature or narrative reviews.

### 2.3. Information Sources, Search Tactics and Further Searches for Primary Studies

An electronic search across the four databases PubMed, Cochrane Database, Scielo, and Scopus was completed on 20 August 2023. We also searched the gray literature using OpenGrey and Google Scholar. The reference lists of the included studies were also examined. Duplicate articles were removed from the identified articles and exported to Zotero^®^, a reference management program developed by the Center for History and New Media in Fairfax County, Virginia, USA. Table 1 lists the search approach used for each database. 

### 2.4. Data Management and Selection Process

The identified articles were inserted into Rayyan^®^ Online Software (Qatar Research Institute of Computing at Hamad Bin Khalifa University (HBKU), Doha, Qatar). The studies were selected in two phases: In phase 1, two reviewers (F.C.O. and E.I.) independently selected the studies by reading the titles and abstracts. Subsequently, phase 2 was carried out, which consisted of reading the full texts, carried out independently by the same two reviewers. A third reviewer (F.C.Z.) was consulted in cases of disagreement.

### 2.5. Data Collection Process

Data from the study were independently collected in duplicate using a table previously prepared by two reviewers (E.B. and W.T.). Data were cross-checked, and any disagreements were resolved by the third review author (P.H.). Information on the following was extracted from the selected articles: authors, year of publication, study design, design of primary studies included, number of studies included in the qualitative and quantitative analysis, country, type of diabetes, results, main conclusions, and used or execution of Meta-analysis, PRISMA, PROSPERO, and Grading of Recommendations Assessment, Development and Evaluation (GRADE).

### 2.6. Evaluation of Meta-Bias, Evidence Quality, and Methodological Quality

Two reviewers (O.S. and R.O.) independently assessed the methodological quality of the included SRs in duplicate using the AMSTAR-2 checklist (A MeaSurement Tool to Assess Systematic Reviews), which was calibrated to a Kappa value of 0.85 [16]. Based on 16 questions that have three alternative answers—”yes”, “no”, or “partially yes”—the AM-STAR-2 assesses the methodological quality of the SR. Classification of a study as having high, moderate, low, or critically low confidence was conducted according to Shea et al. [16].

### 2.7. Measurement Summary

In cases of an SR meta-analysis, we considered the findings of that study along with any results displayed as mean difference, normalized mean difference, relative risk, or odds ratio.

### 2.8. Results Summary

The primary findings of the included SRs were compiled and categorized into the following categories: number of teeth, clinical attachment level, prevalence, general association, plaque index, gingival index, probing depth, and bleeding on probing.

## 3. Results

### 3.1. Examining and Choosing Original Research

After removing duplicates, 519 references remained of the 572 originally yielded from the electronic database search. Phase 1 involved evaluating the titles and abstracts of the selected studies and taking into consideration 12 papers that could be read in full. After adding two more articles from other umbrella reviews and four articles for various reasons, only eighteen SRs remained for quality synthesis. Table 2 provides the rationale behind exclusion for each relevant article [17,18,19,20,21,22,23]. Figure 1 depicts the entire procedure for identification and selection for the research.

### 3.2. Review and Characteristics of Included Studies

The included SRs were published in English between 2009 and 2023. They were carried out in Portugal [10], Peru [24], Malaysia [25], China [26,27], Germany [3], Italy [28,29,30], Australia [31], Japan [32], Denmark [33], the Netherlands [34], Spain [35], Brazil [36,37], and the United States [38,39]. More information on the characteristics of these SRs can be found in Table 3.

### 3.3. Assessment of Methodological Quality and Quality of Evidence

Based on the analysis, 13 SRs [3,10,24,25,26,30,33,34,35,36,37,38,39] were classified as having high confidence, 1 SR [32] as having low confidence, and 4 SRs [27,28,29,31] as having critically low confidence (Table 4).

### 3.4. Overlapping

A total of 296 primary studies [40,41,42,43,44,45,46,47,48,49,50,51,52,53,54,55,56,57,58,59,60,61,62,63,64,65,66,67,68,69,70,71,72,73,74,75,76,77,78,79,80,81,82,83,84,85,86,87,88,89,90,91,92] were identified within the SRs. Of these, 17.26% of the primary studies were included in more than one SR. Thirty-five studies were included twice, eight were included three times, and seven were included four times. More information on the overlap and characteristics of the primary studies is available in Table 5.

### 3.5. Synthesis of Results

The syntheses of the results are presented in Table 3.

### 3.6. General Association

Eleven of the included SRs [3,10,24,27,28,30,31,33,34,38,39] reported an association between PD and DM, while two SRs [32,36] reported no association. Meta-analysis was conducted in six SRs [3,28,33,34,36,38], where the relative risk ratio was found to range from 1.26 (CI: 1.12 to 1.41) [3] to 1.86 (CI: 1.25 to 2.77) [33] and the odds ratio from 1.69 (CI: 0.68 to 4.21) [36] to 2.59 (CI: 2.12 to 3.15) [34]. Costa et al. [10], León-Ríos et al. [24], Nguyen et al. [31], Graziani et al. [30], Ismail et al. [27] and Borgnakke et al. [39] presented the results descriptively and reported an association between PD and DM, while Tanaka et al. [32] reported no association between PD and DM.

### 3.7. Plaque Index

Three included SRs [25,26,29] that conducted meta-analysis reported an association between PD and DM, with a mean difference of 0.20 (CI: 0.18 to 0.23) [26] and a standardized mean difference ranging from 054 (CI: 0.20 to 0.87) [25] 0.71 (CI: 0.19 to 1.22) [29].

### 3.8. Gingival Index

Two included SRs [25,29] that conducted meta-analysis reported an association be-tween PD and DM, with a standardized mean difference ranging from 0.46 (CI: 0.08 to 0.84) [29] to 0.63 (CI: 0.39 to 0.87) [25].

### 3.9. Clinical Attachment Level

Four included SRs [25,26,28,29] reported an association between PD and DM, while one SR [37] reported an association of PD with type 2 diabetes mellitus (T2DM) but not type 1 diabetes mellitus (T1DM). Meta-analysis was conducted in all cases, finding a mean difference ranging from 0.26 (CI: 0.00 to 0.53) to 1.00 (CI: 0.15 to 1.84) [37] and a standardized mean difference ranging from 0.47 (CI: 0.37 to 0.57) [28] to 0.82 (CI: 0.59 to 1.04) [29].

### 3.10. Number of Teeth

One included SR [26] that conducted meta-analysis reported an association between PD and DM, with a mean difference of −2.14 (CI: −2.87 to −1.40).

### 3.11. Prevalence

Three included SRs [26,28,35] reported an association between PD and DM. Me-ta-analysis was conducted in two of these [26,28], where they found the odds ratio ranged from 0.19 (CI: 0.08 to 0.37) [28] to 1.85 (CI: 1.61 to 2.11) [26]. Mauri-Obradors et al. [35] presented the results descriptively and reported an association between PD and DM.

### 3.12. Probing Depth

Three included SRs [25,26,29] reported an association between PD and DM, while one SR [37] reported that an association of PD with T2DM but not T1DM. All conducted meta-analysis and found a mean difference ranging from 0.11 (CI: −0.03 to 0.25) to 0.46 (CI: 0.01 to 0.91) [37] and a standardized mean difference ranging from 0.36 (CI: 0.16 to 0.55) [29] to 0.67 (CI: 0.23 to 1.11) [25].

### 3.13. Bleeding on Probing

Three included SRs [25,26,29] reported an association between PD and DM. Two studies conducted meta-analysis, finding a mean difference of 7.90 (CI: 4.24 to 11.56) [26] and a standardized mean difference ranging from 0.32 (CI: 0.07 to 0.58) [25] to 0.65 (CI: 0.08 to 1.23) [29].

## 4. Discussion

The evaluation and analysis of the association between PD and systemic diseases, such as DM, has been of interest in recent years. Numerous RCTs have investigated this topic and reported supporting evidence for an association between these two diseases.

Some studies have evaluated this association in a general way, while others have evaluated it according to periodontal clinical parameters. As there are now several published SRs analyzing the association between PD and DM, it has become necessary to compile data from these to assess the methodological quality of each study.

Currently, three relevant umbrella reviews have been conducted, and all [11,93,94] reviewed the most current evidence on the nature of the relationship/association between PD and DM; however, studies that analyzed whether periodontal therapy was effective in people with DM were included, so the results and conclusions must be considered with caution.

An exhaustive literature search was conducted in the present study to summarize the available SRs investigating the association between PD and DM (T1DM, T2DM, and GDM), identifying 18 SRs that matched the inclusion criteria and were included in the analysis. SRs represent the highest level of the scientific evidence pyramid, but their results should be evaluated cautiously due to the potential for bias.

The SRs included in this study exhibit certain limitations related to the selected primary studies, such as the inclusion of different of study types, the selection criteria for inclusion of studies, the inclusion of different population groups (children, adolescents, and adults), different diagnostic criteria for periodontal diseases, and the evaluation of different types of diabetes mellitus (T1DM, T2DM and GDM).

On the other hand, more than 50% of the included studies demonstrated a high level of confidence, potentially enhancing the level of evidence for the results and conclusions presented in this study. However, the continued publication of SRs lacking a high level of confidence highlights the need for greater rigor in the development of such studies on this topic.

The AMSTAR-2 instrument, in its most current version, was used to assess the methodological quality of the included SRs. A factor related to the methodological quality that deserves emphasis is the critical domains 2, 4, 7, 9, 13, and 15 of AMSTAR-2. Some SRs failed to explicitly assess their methods, did not use an exhaustive search strategy, did not provide justification for the exclusion of studies, did not use a satisfactory technique to assess the risk of bias, and did not consider the risk of bias of the included studies when interpreting or discussing their results. In addition, they did not report the risk of publication bias. This highlights a need for these elements to be included in the development of future SRs.

Additionally, caution should be taken as some studies were included in the SRs more than once, leading to data being reevaluated multiple times, which could skew perceptions and findings. Although conducting new SRs to overcome methodological limitations, as recommended by Moher [95], would be of interest, in consideration of the high overlap rate, it is more important to carry out well-conducted RCTs with long-term follow-up and from different research groups to expand knowledge on this topic.

### 4.1. Evidence Summary

The present umbrella review was carried out to advance research on the association between PD and DM, with the aim of minimizing biases and random errors in SRs and meta-analyses on this subject. The findings are summarized and discussed below while considering the limitations of the SRs used in this study.

The SRs included in this study support a general association between PD and DM. This is similar to the report by the Consensus Report of the European Federation of Periodontics [17], which highlights that periodontitis is associated with diabetes mellitus. However, two of the included studies [32,36] reported no association between these two diseases, where one [32] focused only on the Japanese population, concluding there was little evidence on this topic for this population, while the other [36] performed subgroup analysis according to type of primary study, concluding that the association is supported by the cross-sectional studies but not the case–control studies.

In terms of plaque index, gingival index, number of missing teeth, prevalence, and bleeding on probing, the studies indicate that patients with DM have a greater probability of increases in all these clinical parameters, indicating an association between PD and DM.

The studies also indicate that patients with DM have a greater probability of increases in all these clinical parameters when considering the depth of catheterization and the clinical attachment level. However, one study [37] indicated that PD is associated with T2DM but not T1DM.

Furthermore, the following should be noted regarding the included studies: four [10,27,28,29] analyzed T1DM, with only one [10] demonstrating high overall confidence; three [24,36,38] analyzed GDM, with all demonstrating high overall confidence; and ten [3,25,26,30,31,32,33,35,37,39] analyzed DM in general, with eight [3,25,26,30,33,35,37,39] classified as having high general confidence.

### 4.2. Implications for Clinical Practice

Given the association between PD and DM, accurate anamnesis and correct diagnosis are critical to ensure that patients are presented with the most viable, straightforward, and least invasive treatment options, such as scaling and root planning, or the use of lasers and local antimicrobials.

### 4.3. Implications for Research

There are a high number of SRs available, but it is evident from our assessment that there is an immediate need for improved reporting quality. For future SRs, we recommend examining the instruments used in guiding their development, including quality evaluation tools. To obtain more reliable results, additional SRs on the relationship between PD and MGD should be conducted with strong methodological rigor. For future primary studies, we also recommend using the latest definitions for the diagnosing periodontal diseases.

## 5. Conclusions

According to the findings of this umbrella review, there is strong evidence supporting the association between periodontal disease and various forms of diabetes mellitus, including type 1 and type 2, and gestational diabetes. The reviewed studies demonstrate a consistent relationship across different clinical parameters, such as plaque index, gingival index, clinical attachment level, probing depth, and bleeding on probing, with diabetes patients showing a higher likelihood of experiencing worsened periodontal conditions. However, two studies did not find an association, highlighting the need for further research to address potential differences in study populations and methodologies.

## Figures and Tables

**Figure 1 healthcare-12-02311-f001:**
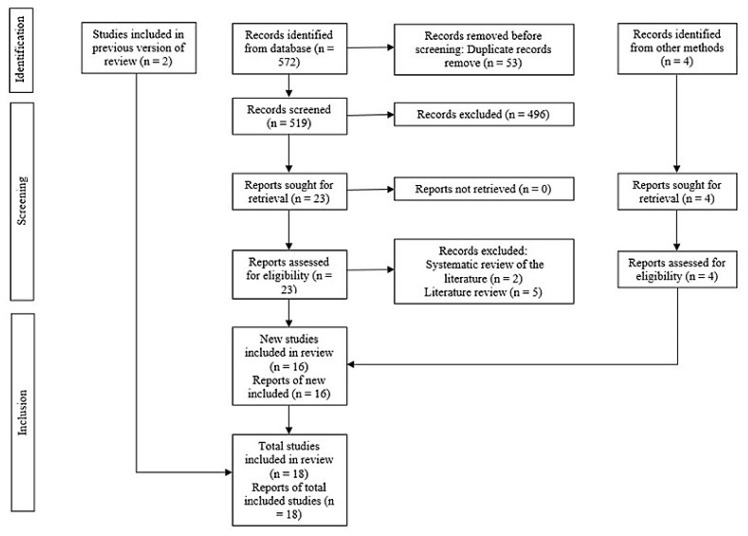
PRISMA diagram showing the process for inclusion and exclusion of studies.

**Table 1 healthcare-12-02311-t001:** Database search strategy.

Database	Search Strategy	Number of Studies
PubMed	((“Periodontal disease”) OR (“gingivitis”) OR (“periodontitis”)) AND ((“Diabetes mellitus”) OR (“DM1”) OR (“DM2”) OR (“gestational diabetes”) OR (“type 1 diabetes”) OR (“type 2 diabetes”)) AND (“association”) AND ((“systematic review”) OR (“meta-analysis”) OR (“systematic review and meta-analysis”))	66
Cochrane Database	#1 MeSH descriptor: [Periodontal Diseases] explode all trees #2 MeSH descriptor: [Periodontitis] in all MeSH products #3 MeSH descriptor: [Gingivitis] explode all trees #4 (“Periodontal disease”): ti, ab, kw OR (“gingivitis”): ti, ab, kw OR (“periodontitis”): ti, ab, kw #5 #1 OR #2 OR #3 OR #4 #6 MeSH descriptor: [Diabetes Mellitus] explode all trees #7 MeSH descriptor: [Diabetes Mellitus, Type 2] explode all trees #8 MeSH descriptor: [Diabetes Mellitus, Type 1] explode all trees #9 MeSH descriptor: [Diabetes, Gestational] explode all trees #10 (“Diabetes mellitus”): ti, ab, kw OR (“DM1”): ti, ab, kw OR (“DM2”): ti, ab, kw OR (“gestational diabetes”): ti, ab, kw OR (“type 1 diabetes mellitus”): ti, ab, kw #11 (“type 2 diabetes mellitus”): ti, ab, kw #12 #6 OR #7 OR #8 OR #9 OR #10 OR #11 #13 MeSH descriptor: [Association] explode all trees #14 (“association”): ti, ab, kw #15 #13 OR #14 #16 #5 AND #12 AND #15	1
Scielo	((“Periodontal disease”) OR (“gingivitis”) OR (“periodontitis”)) AND ((“Diabetes mellitus”) OR (“DM1”) OR (“DM2”) OR (“gestational diabetes”) OR (“type 1 diabetes”) OR (“type 2 diabetes”)) AND (“association”) AND ((“systematic review”) OR (“meta-analysis”) OR (“systematic review and meta-analysis”))	3
Scopus	(TITLE-ABS-KEY (((“Periodontal disease”) OR (“gingivitis”) OR (“periodontitis”))) AND TITLE-ABS-KEY (((“Diabetes mellitus”) OR (“DM1”) OR (“DM2”) OR (“gestational diabetes”) OR (“type 1 diabetes”) OR (“type 2 diabetes”))) AND TITLE-ABS-KEY ((“association”)) AND TITLE-ABS-KEY (((“systematic review”) OR (“meta-analysis”) OR (“systematic review and meta-analysis”)))) AND (LIMIT-TO (SRCTYPE, “j”)) AND (LIMIT-TO (DOCTYPE, “re”))	92
Google Scholar	“Periodontal disease” + “Diabetes mellitus” OR “gestational diabetes” + “association” + “systematic review” − “in vitro” − “literature”	410
OpenGrey	(“Periodontal disease”) AND ((“Diabetes mellitus”) OR (“gestational diabetes”)) AND (“association”) AND (“systematic review”)	0

**Table 2 healthcare-12-02311-t002:** Reasons for exclusion of studies.

Author	Reason for Exclusion
Herrera et al. [17]	Literature review
Li et al. [18]	
George et al. [19]	
Bansal et al. [20]	
Laddha et al. [21]	
Salvi et al. [22]	Systematic review of the literature
Mealey et al. [23]	

**Table 3 healthcare-12-02311-t003:** Characteristics of included studies.

Authors	Year	Study Design	Included Study Designs	Number of Studies in the Qualitative Analysis	Number of Studies in the Quantitative Analysis	Type of Diabetes	Outcomes			Conclusions
Costa et al. [10]	2023	SR	CCs, Cs, CSs and RCTs	15	0	T1DM	Most studies confirm the association between T1DM and PDs.			The prevalence and severity of PD was higher in patients with T1DM compared to healthy subjects.
León-Ríos et al. [24]	2022	SR	CCs, Cs and CSs	8	0	GDM	In most studies an association between PD and GDM was verified.			PD increases the risk of developing GDM.
Zainal et al. [25]	2021	SR with MA	CCs and CSs	11	11	DM	PI		SMD = 0.54 (0.20–0.87)	The duration of DM in children and adolescents affects periodontal status even with a difference of 1 year. Thus, children and adolescents with DM and early PD will progress to periodontitis without intervention.
							GI		SMD = 0.63 (0.39–0.87)	
							BP		SMD = 0.32 (0.07–0.58)	
							PDT		SMD = 0.67 (0.23–1.11)	
							CAL		SMD = 0.79 (0.52–1.05)	
Zheng et al. [26]	2021	SR with MA	CCs, Cs and CSs	40	27	DM	P		OR = 1.85 (1.61–2.11)	The prevalence and severity of periodontitis are higher in patients with diabetes than in non-diabetic populations.
							PDT		MD = 0.23 (0.17–0.29)	
							PI		MD = 0.20 (0.18–0.23)	
							CAL		MD = 0.39 (0.28–0.50)	
							NT		MD = −2.14 (−2.87–1.40)	
							BP		MD = 7.90 (4.24–11.56)	
Stöhr et al. [3]	2021	SR with MA	Cs	15	15	DM	GA		RR = 1.26 (1.12–1.41)	The findings show a positive bidirectional association between PD and DM.
Dicembrini et al. [28]	2020	SR with MA	CCs, Cs and CSs	19	19	T1DM	P		OR = 0.19 (0.08–0.37)	The data confirm that T1DM is a relevant risk factor for the development of PD.
							GA		OR = 2.52 (1.33–4.76)	
							CAL		SMD = 0.47 (0.37–0.57)	
Nguyen et al. [31]	2020	SR	Ls, CCs and CSs	14	0	DM	Higher risks of diabetic complications have been reported in people with diabetes and periodontitis compared to those with diabetes who do not have periodontitis.			This review systematically addressed current epidemiological data providing evidence that periodontitis is associated with an increased risk of developing diabetic complications compared to patients without periodontitis.
Rapone et al. [29]	2020	SR with MA	CCs and CSs	10	10	T1DM	BP		SMD = 0.65 (0.08–1.23)	Although evidence suggests a possible association between PD and T1DM in children and adolescents, the study designs and methodological limitations make interpretation difficult.
							CAL		SMD = 0.82 (0.59–1.04)	
							GI		SMD = 0.46 (0.08–0.84)	
							PI		SMD = 0.71 (0.19–1.22)	
							PDT		SMD = 0.36 (0.16–0.55)	
Tanaka et al. [32]	2019	SR	SRs, Cs, CCs and CSs	33	0	DM	Several cohort studies and MAs found epidemiological evidence for cancer development in diabetes but little for other conditions.			In Japan, there is little evidence that diabetes affects PD incidence and prevalence.
Nascimento et al. [33]	2018	SR with MA	Ls	13	6	DM	GA		RR = 1.86 (1.25–2.77)	This study provides evidence that diabetes is associated with an increased risk of periodontitis onset and progression in adults.
Ziukaite et al. [34]	2018	SR with MA	CCs, Cs and CSs	29	21	DM	GA		OR = 2.59 (2.12–3.15)	Higher overall prevalence and odds of having diabetes in populations with periodontitis compared to without.
Graziani et al. [30]	2018	SR	CCs, Cs and CSs	20	0	DM	Healthy individuals with periodontitis have poor glycemic control and an increased risk of developing diabetes. People affected by diabetes show impaired glycemic control and a significantly higher prevalence of diabetes-related complications if they also suffer from periodontitis. There is limited evidence available on GDM and T1DM.			Periodontitis has a significant impact on diabetes control, incidence, and complications.
Mauri-Obradors et al. [35]	2017	SR	Ls and CSs	19	0	DM	PD was more prevalent among diabetic patients.			There are multiple oral manifestations associated with DM, including PD.
Lima et al. [36]	2016	SR with MA	CCs and CSs	8	7	GDM	GA	CSs	OR = 1.67 (1.20–2.32)	GA
								CCs	OR = 1.69 (0.68–4.21)	
Abariga et al. [38]	2016	SR with MA	CCs, Cs and CSs	10	10	GDM	GA		OR = 2.08 (1.21–3.58)	The MAs suggest a statistically significant increased risk of GDM in women with periodontitis compared to without.
Ismail et al. [27]	2015	SR	CCs and Ls	28	0	T1DM	Most studies reported significantly greater plaque accumulation and higher GI in children with T1DM. Cohort studies reported no significant differences in periodontal parameters over time.			There is evidence that children with T1DM show poorer periodontal health with greater plaque accumulation compared to healthy children.
Borgnakke et al. [39]	2013	SR	CCs, Cs, CSs, Ls and Rs	16	0	DM	Some supporting evidence for PD having significant adverse effects on glycemic control, diabetes complications, and the development of T2DM and possibly GDM.			Current evidence suggests that PD negatively affects diabetes outcomes.
Chávarry et al. [37]	2009	SR with MA	Ls and CSs	57	16	DM	CAL	T1DM	MD = 0.26 (−0.00–0.53)	T2DM can be considered a risk factor for periodontitis. More studies are needed to confirm the harmful effects of T1DM on PD.
								T2DM	MD = 1.00 (0.15–1.84)	
							PDT	T1DM	MD = 0.11 (−0.03–0.25)	
								T2DM	MD = 0.46 (0.01–0.91)	

SR = systematic review; MA = meta-analysis; CC = case and control; C = cohort; CS = cross-sectional; RCT = randomized clinical trial; L = longitudinal study; R = retrospective study; T1DM = type 1 diabetes mellitus; T2DM = type 2 diabetes mellitus; GDM = gestational diabetes mellitus; PD = periodontal disease; DM = diabetes mellitus; PI = plaque index; GI = gingival index; BP = bleeding on probing; PDT = probing depth; CAL = clinical attachment level; P = prevalence; NT = number of teeth; GA = general association; RR = risk ratio; MD = mean difference; OR = odds ratio; SMD = standardized mean difference.

**Table 4 healthcare-12-02311-t004:** Assessment of the methodological quality and quality of evidence of the included studies.

Authors	Year	AMSTAR–2																Overall Confidence
		1	2 *	3	4 *	5	6	7 *	8	9 *	10	11 *	12	13 *	14	15 *	16	
Costa et al. [10]	2023	Y	Y	Y	Yes partial	Y	N	Yes partial	Y	Y	Y	No meta-analysis		Y	Y	No meta-analysis	Y	High
León-Ríos et al. [24]	2022	Y	Yes partial	Y	Yes partial	Y	N	Yes partial	Y	Y	Y	No meta-analysis		Y	Y	No meta-analysis	Y	High
Zainal et al. [25]	2021	Y	Y	Y	Yes partial	N	Y	Yes partial	Y	Y	Y	Y	Y	Y	Y	Y	Y	High
Zheng et al. [26]	2021	Y	Y	Y	Yes partial	Y	Y	Yes partial	Y	Y	Y	Y	Y	Y	Y	Y	Y	High
Stöhr et al. [3]	2021	Y	Y	Y	Yes partial	Y	Y	Y	Y	Y	Y	Y	Y	Y	Y	Y	Y	High
Dicembrini et al. [28]	2020	Y	Yes partial	Y	Yes partial	Y	Y	N	Y	Y	Y	Y	Y	Y	Y	N	Y	Critically low
Nguyen et al. [31]	2020	Y	N	Y	N	Y	Y	N	Y	Y	Y	No meta-analysis		Y	Y	No meta-analysis	Y	Critically low
Rapone et al. [29]	2020	Y	Yes partial	Y	N	Y	Y	N	Y	Y	Y	Y	Y	Y	Y	N	Y	Critically low
Tanaka et al. [32]	2019	Y	Y	Y	Yes partial	Y	Y	N	Y	Y	Y	No meta-analysis		Y	Y	No meta-analysis	N	Low
Nascimento et al. [33]	2018	Y	Y	Y	Yes partial	Y	Y	Y	Y	Y	Y		Y	Y	Y	Y	Y	High
Ziukaite et al. [34]	2018	Y	Yes partial	Y	Yes partial	Y	Y	Yes partial	Y	Y	Y	Y	Y	Y	Y	Y	Y	High
Graziani et al. [30]	2018	Y	Yes partial	Y	Yes partial	Y	Y	Yes partial	Y	Y	Y	No meta-analysis		Y	Y	No meta-analysis	Y	High
Mauri-Obradors et al. [35]	2017	Y	Yes partial	Y	Y	Y	Y	Yes partial	Y	Y	Y	No meta-analysis		Y	Y	No meta-analysis	Y	High
Lima et al. [36]	2016	Y	Y	Y	Yes partial	Y	Y	Y	Y	Y	Y	Y	Y	Y	Y	Y	N	High
Abariga et al. [38]	2016	Y	Yes partial	Y	Yes partial	N	Y	Yes partial	Y	Y	Y	Y	Y	Y	Y	Y	Y	High
Ismail et al. [27]	2015	Y	N	Y	Yes partial	Y	Y	Yes partial	Y	Y	Y	No meta-analysis		N	Y	No meta-analysis	Y	Critically low
Borgnakke et al. [39]	2013	Y	Yes partial	Y	Yes partial	Y	Y	Yes partial	Y	Y	Y	No meta-analysis		Y	Y	No meta-analysis	Y	High
Chávarry et al. [37]	2009	Y	Yes partial	Y	Yes partial	Y	Y	Yes partial	Y	Y	Y	Y	Y	Y	Y	Y	N	High

AMSTAR = A MeaSurement Tool to Assess Systematic Reviews. Y = Yes N = No 1 = Did the research questions and inclusion criteria for the review include components of PICO? 2 = Did the review report contain an explicit statement that the review methods were established prior to the conduct of the review and did it justify any significant deviations from the protocol? 3 = Did the review authors explain their selection of the study designs for inclusion in the review? 4 = Did the review authors use a comprehensive literature search strategy? 5 = Did the review authors perform study selection in duplicate? 6 = Did the review authors perform data extraction in duplicate? 7 = Did the review authors provide a list of excluded studies and justify the exclusions? 8 = Did the review authors describe the included studies in adequate detail? 9 = Did the review authors use a satisfactory technique for assessing the risk of bias (RoB) in individual studies included in the review? 10 = Did the review authors report on the sources of funding for the studies included in the review? 11 = If meta-analysis was performed, did the review authors use appropriate methods for statistical combination of results? 12 = If meta-analysis was performed, did the review authors assess the potential impact of RoB in individual studies on the results of the meta-analysis or other evidence synthesis? 13 = Did the review authors account for RoB in primary studies when interpreting/discussing the review results? 14 = Did the review authors provide a satisfactory explanation for, and discussion of, any heterogeneity observed in the review results? 15 = If they performed quantitative synthesis, did the review authors carry out an adequate investigation of publication bias (small study bias) and discuss its likely impact on the review results? 16 = Did the review authors report any potential sources of conflicts of interest, including any funding they received for conducting the review? * = Critical domain.

**Table 5 healthcare-12-02311-t005:** Overlapping of primary studies in systematic reviews.

Primary Studies	Systematic Reviews That Included the Primary Studies	Times That Primary Studies Were Included
Bullon et al. [40]	Zheng et al. [26], Graziani et al. [30], Lima et al. [36], Abariga et al. [38]	4
Al-Khabbaz et al. [41]	Zainal et al. [25], Dicembrini et al. [28], Rapone et al. [29], Ismail et al. [27]	4
Esteves Lima et al. [42]	León-Ríos et al. [24], Graziani et al. [30], Lima et al. [36], Abariga et al. [38]	4
Chokwiriyachit et al. [43]	Zheng et al. [26], Graziani et al. [30], Lima et al. [36], Abariga et al. [38]	4
Morita et al. [44]	Stöhr et al. [3], Tanaka et al. [32], Nascimento et al. [33], Borgnakke et al. [39]	4
Xiong et al. [45]	Zheng et al. [26], Lima et al. [36], Abariga et al. [38], Borgnakke et al. [39]	4
Dakovic et al. [46]	Zainal et al. [25], Dicembrini et al. [28], Rapone et al. [29], Ismail et al. [27]	4
Dasanayake et al. [47]	Zheng et al. [26], Lima et al. [36], Abariga et al. [38], Borgnakke et al. [39]	4
Saito et al. [48]	Zheng et al. [26], Tanaka et al. [32], Borgnakke et al. [39], Chávarry et al. [37]	4
Sun et al. [49]	Costa et al. [10], Stöhr et al. [3], Dicembrini et al. [28]	3
Ismail et al. [50]	Costa et al. [10], Zainal et al. [25], Rapone et al. [29]	3
Chiu et al. [51]	Stöhr et al. [3], Nascimento et al. [33], Graziani et al. [30]	3
Habib [52]	Zheng et al. [26], Lima et al. [36], Abariga et al. [38]	3
Orbak et al. [53]	Zainal et al. [25], Rapone et al. [29], Ismail et al. [27]	3
Lalla et al. [54]	Zainal et al. [25], Rapone et al. [29], Ismail et al. [27]	3
Aren et al. [55]	Zainal et al. [25], Ismail et al. [27], Chávarry et al. [37]	3
Sbordone et al. [56]	Nascimento et al. [33], Ismail et al. [27], Chávarry et al. [37]	3
Firatli [57]	Nascimento et al. [33], Ismail et al. [27], Chávarry et al. [37]	3
Pinson et al. [58]	Dicembrini et al. [28], Ismail et al. [27], Chávarry et al. [37]	3
Roy et al. [59]	Costa et al. [10], Dicembrini et al. [28]	2
Babu et al. [60]	Zainal et al. [25], Rapone et al. [29]	2
Myllymäki et al. [61]	Zheng et al. [26], Stöhr et al. [3]	2
Chaparro et al. [62]	León-Ríos et al. [24], Zheng et al. [26]	2
Winning et al. [63]	Zheng et al. [26], Stöhr et al. [3]	2
Jindal et al. [64]	Costa et al. [10], Dicembrini et al. [28]	2
Popławska-Kita et al. [65]	Costa et al. [10], Dicembrini et al. [28]	2
Amiri et al. [66]	Nguyen et al. [31], Graziani et al. [30]	2
Lee et al. [67]	Nascimento et al. [33], Ziukaite et al. [34]	2
Jimenez et al. [68]	Stöhr et al. [3], Nascimento et al. [33]	2
Southerland et al. [69]	Nguyen et al. [31], Borgnakke et al. [39]	2
Hodge et al. [70]	Zheng et al. [26], Dicembrini et al. [28]	2
Ruiz et al. [71]	Lima et al. [36], Abariga et al. [38]	2
Ide et al. [72]	Stöhr et al. [3], Borgnakke et al. [39]	2
Tagelsir et al. [73]	Zainal et al. [25], Ismail et al. [27]	2
Abrao et al. [74]	Nguyen et al. [31], Borgnakke et al. [39]	2
Demmer et al. [75]	Stöhr et al. [3], Borgnakke et al. [39]	2
Shultis et al. [76]	Nguyen et al. [31], Borgnakke et al. [39]	2
Novak et al. [77]	Lima et al. [36], Abariga et al. [38]	2
Borges-Yáñez [78]	Ziukaite et al. [34], Chávarry et al. [37]	2
Mansour et al. [79]	Zheng et al. [26], Chávarry et al. [37]	2
Campus et al. [80]	Zheng et al. [26], Chávarry et al. [37]	2
Saremi et al. [81]	Nguyen et al. [31], Borgnakke et al. [39]	2
Siudikiene et al. [82]	Zainal et al. [25], Ismail et al. [27]	2
Noma et al. [83]	Nguyen et al. [31], Borgnakke et al. [39]	2
Marugame et al. [84]	Tanaka et al. [32], Chávarry et al. [37]	2
Zielinski et al. [85]	Zheng et al. [26], Chávarry et al. [37]	2
Kawamura et al. [86]	Zheng et al. [26], Chávarry et al. [37]	2
Collin et al. [87]	Zheng et al. [26], Chávarry et al. [37]	2
Firatli et al. [88]	Ismail et al. [27], Chávarry et al. [37]	2
Thorstensson et al. [89]	Nguyen et al. [31], Borgnakke et al. [39]	2
Sbordone et al. [90]	Ismail et al. [27], Chávarry et al. [37]	2
de Pommereau et al. [91]	Ismail et al. [27], Chávarry et al. [37]	2
Sandholm et al. [92]	Ismail et al. [27], Chávarry et al. [37]	2

## Data Availability

Data sharing is not applicable. No new data were created or analyzed in this study.

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
