# Peer review of "An Umbrella Review of the Association Between Periodontal Disease and Diabetes Mellitus"

_healthcare, 2024, doi:10.3390/healthcare12222311_

Round 1

Reviewer 1 Report

Comments and Suggestions for Authors

This is an excellent topic for drafting an umbrella review. The review was conducted in a scientifically rigorous manner and clearly outlined the inclusion and exclusion criteria. However, it needs significant grammatical revisions. Many sentences are overly complex and have been paraphrased inaccurately, which can make some conclusions hard to review critically. Unnecessary filler words that appear throughout the article need to be removed. The limitations of the study should be clearly defined.

L22-24; L28-29: Grammatical modification necessary.

L31-32: Please be specific with the confidence level for asserting the conclusion.

L34: Please also include keywords that are not present in the title.

L40-41: Reference 3 does indicate the first part of the sentence, but it doesn't say if DM will be one of the five main causes of death. Please specify a proper reference or modify the statements.

L48: “20 and 60% of the world’s population”. This seems to be a quite large range. Please review more literature for better specificity.

L61-63: I'm not sure if this is correct. Please refer: PMID: 33643418

L85: Modify it to: An electronic search across four databases (Pubmed, Cochrane database, Scielo, and Scopus) was completed on August 20, 2023.

L113-114: How was the confidence level quantitatively measured?

P2L17: Remove ‘they’.

P3L60: Please specify some of the periodontal clinical parameters here as well for clarity.

P3L64-67: Grammatical modification necessary.

P3L73: Why is it assumed that the results of SRs are likely biased?

P3L85-92: This sentence needs to be concise and clear. What factor are we talking about?

P4L95: Did you mean that the perception can become unclear due to multiple re-evaluations?

P4L107: Replace ‘point out’ with ‘specify’.

P4l129: It will be useful to mention here about some common minimally invasive methods of periodontal disease treatment that can be used instead of more common invasive methods.

Comments on the Quality of English Language

Major grammatical modifications are necessary. Unnecessary filler words need to be removed. The length of the sentences needs to be significantly reduced in many paragraphs.

Author Response

Reviewer 1

Comments 1: This is an excellent topic for drafting an umbrella review. The review was conducted in a scientifically rigorous manner and clearly outlined the inclusion and exclusion criteria. However, it needs significant grammatical revisions. Many sentences are overly complex and have been paraphrased inaccurately, which can make some conclusions hard to review critically. Unnecessary filler words that appear throughout the article need to be removed. The limitations of the study should be clearly defined.

Response 1: We appreciate the positive feedback on our topic selection and methodology. In response, we used MDPI Author Services to refine grammar and simplify complex sentences. Additionally, we have clarified the study limitations, as noted in L69-72 and L88-93: “The SRs included in this study exhibit certain limitations related to the selected primary studies, such as the inclusion of different types of studies, criteria for selection, and population groups…”. We have further discussed the potential for multiple data re-evaluations, as highlighted.

Comments 2: L22-24; L28-29: Grammatical modification necessary.

Response 2: We revised this section with support from MDPI Author Services. The revised text is: “We included systematic reviews (SRs) with or without meta-analysis evaluating primary studies that investigated the association between periodontal disease and diabetes mellitus, and there were no time or language restrictions.

Comments 3: L31-32: Please be specific with the confidence level for asserting the conclusion.

Response 3: Revised with assistance from MDPI Author Services to: “Conclusion: The findings and conclusions of this umbrella review indicate with high confidence that periodontal disease is associated with the onset of type 1 and type 2 diabetes mellitus and gestational diabetes.

Comments 4: L34: Please also include keywords that are not present in the title.

Response 4: Additional keywords such as “gestational diabetes,” “systematic review,” and “meta-analysis” have been included.

Comments 5: L40-41: Reference 3 does indicate the first part of the sentence, but it doesn't say if DM will be one of the five main causes of death. Please specify a proper reference or modify the statements.

Response 5: We have corrected the sentence with assistance from MDPI Author Services: “It is estimated that by 2045, there will be around 700 million people living with diabetes, which will be one of the five main causes of death [3].

Comments 6: L48: “20 and 60% of the world’s population”. This seems to be a quite large range. Please review more literature for better specificity.

Response 6: We reviewed additional sources and verified the range, which is accurate. The revised sentence reads: “It is estimated that between 20 and 60% of the world’s population suffers from periodontal disease [7,8].

Comments 7: L61-63: I'm not sure if this is correct. Please refer: PMID: 33643418.

Response 7: The statement was reviewed, and we retained it as originally phrased: “To date, no comprehensive synthesis or evaluation of all systematic reviews, including those from recent years, has been performed.

Comments 8: L85: Modify it to: An electronic search across four databases (Pubmed, Cochrane database, Scielo, and Scopus) was completed on August 20, 2023.

Response 8: Revised as suggested. The updated text now states: “An electronic search across the four databases PubMed, Cochrane Database, Scielo, and Scopus was completed on August 20, 2023.

Comments 9: L113-114: How was the confidence level quantitatively measured?

Response 9: In response to your question about the quantitative measurement of the confidence level, the following clarification was made with support from MDPI Author Services: “Classification of a study as having high, moderate, low, or critically low confidence was conducted according to Shea et al. [16].” This phrasing should be interpreted as an assessment that did not involve quantitative measurement. Instead, it reflects a qualitative categorization, where the studies included in the review were rated based on the number of studies at each confidence level: high, moderate, low, and critically low.

Comments 10: P2L17: Remove ‘they’.

Response 10: Revised as suggested. The corrected text reads: “and Borgnakke et al. [39] presented the results descriptively and reported an association between PD and DM, while Tanaka et al. [32] reported no association.

Comments 11: P3L60: Please specify some of the periodontal clinical parameters here as well for clarity.

Response 11: We have included specific clinical parameters to improve clarity: “Some studies have evaluated this association in a general way, while others have evaluated it according to periodontal clinical parameters.”

Comments 12: P3L64-67: Grammatical modification necessary.

Response 12: Revised for clarity with support from MDPI Author Services: “Currently, three relevant umbrella reviews have been conducted, and all [11,93,94] reviewed the most current evidence on the nature of the relationship/association between PD and DM; however, studies that analyzed whether periodontal therapy was effective in people with DM were included, so the results and conclusions must be considered with caution.

Comments 13: P3L73: Why is it assumed that the results of SRs are likely biased?

Response 13: We revised the statement regarding bias potential with assistance from MDPI Author Services. The corrected text now reads: “SRs represent the highest level of the scientific evidence pyramid, but their results should be evaluated cautiously due to the potential for bias.” This clarification aims to indicate that while the results themselves may not be inherently biased, the included studies did not consistently meet high-confidence standards, which introduces a possibility of bias.

Comments 14: P3L85-92: This sentence needs to be concise and clear. What factor are we talking about?

Response 14: Revised with assistance from MDPI Author Services: “The AMSTAR-2 instrument was used to assess methodological quality. Critical domains such as 2, 4, 7, 9, 13, and 15 revealed limitations in some SRs, including lack of an exhaustive search strategy, insufficient reporting of excluded studies, unsatisfactory risk of bias assessment, and absence of publication bias reporting, highlighting areas for improvement in future SRs.

Comments 15: P4L95: Did you mean that the perception can become unclear due to multiple re-evaluations?

Response 15: The clarification here addresses the reviewer’s question about re-evaluation. With support from MDPI Author Services, we revised the text as follows: “Additionally, caution should be taken as some studies were included in the SRs more than once, leading to data being reevaluated multiple times, which could skew perceptions and findings.” To clarify, the repetition of certain primary studies across multiple SRs could lead to an overestimation of effects, which may affect the interpretation of the results.

Comments 16: P4L107: Replace ‘point out’ with ‘specify’.

Response 16: Revised as suggested. The updated text reads: “This is similar to the report by the Consensus Report of the European Federation of Periodontics [17], which highlights that periodontitis is associated with diabetes mellitus.

Comments 17: P4l129: It will be useful to mention here about some common minimally invasive methods of periodontal disease treatment that can be used instead of more common invasive methods.

Response 17: We have included: “Given the association between PD and DM, accurate anamnesis and correct diagnosis are critical to ensure that patients are presented with the most viable, straightforward, and least invasive treatment options, such as scaling and root planning, or the use of lasers and local antimicrobials.

Reviewer 2 Report

Comments and Suggestions for Authors

1)Item 2.5, Line 106: “Mata-analysis”. Please check

2)Lines 109-111:  “Two reviewers (O.S. and R.O.) independently and in duplicate assessed the methodological quality of the included SRs using the AMSTAR-2 checklist (A MeaSurement Tool 110 to Assess Systemic Reviews), which was calibrated (Kappa 0.85)” [16]. What/who was calibrated?

3)Figure 1: the text in the figure seems a little blurry;

4)Table 4: it could be more clean and clear; Maybe replace “yes” with “Y” and “No” with “N”. Or, instead of using a table, the authors could use a figure, placing the questions in the body of the figure…

5)Numbering of lines and pages has been interrupted in page 8 and then stars again in the item “3.4”

6)Item 3.6, line 11:  “Eleven SRs [3,10,24,27,28,30,31,33,34,38,39] included reported that there were an as”   Please, revise the sentence

7)Item 3.8: “Two SR [25,29] included reported that there is an association between PD and DM. This study meta-analyzed its results, where they found that the standardized mean difference ranged from 0.46 (CI: 0.08 to 0.84) [29] to 0.63 (CI: 0.39 to 0.87) [25]”. Please revise the sentence

8)Discussion, line 87:  “they did not use a search strategy was exhaustive”… Please check this sentence

9)Item 4.1, lines 122-125: Furthermore, it should be noted that of the included studies: four [10,27–29] analyze  T1DM, with only one [10] having high overall confidence; three [24,36,38] analyzing GDM  where all have high overall confidence; and ten [3,25,26,30–33,35,37,39] analyzing DM in general, eight [3,25,26,30,33,35,37,39] of which They have high general confidence.” Please review this sentence

Author Response

Reviewer 2

Comments 1: 1) Item 2.5, Line 106: “Mata-analysis”. Please check

Response 1: The term has been corrected to "Meta-analysis."

Comments 2: 2) Lines 109-111: “Two reviewers (O.S. and R.O.) independently and in duplicate assessed the methodological quality of the included SRs using the AMSTAR-2 checklist (A MeaSurement Tool 110 to Assess Systemic Reviews), which was calibrated (Kappa 0.85)” [16]. What/who was calibrated?

Response 2: The calibration refers to the two reviewers, who calibrated their evaluations to ensure consistency in applying the AMSTAR-2 criteria.

Comments 3: 3) Figure 1: the text in the figure seems a little blurry;

Response 3: The image quality has been improved to ensure clarity.

Comments 4: 4) Table 4: it could be more clean and clear; Maybe replace “yes” with “Y” and “No” with “N”. Or, instead of using a table, the authors could use a figure, placing the questions in the body of the figure…

Response 4: The suggestion has been accepted, and the changes have been made to enhance the table’s readability.

Comments 5: 5) Numbering of lines and pages has been interrupted in page 8 and then stars again in the item “3.4”

Response 5: The formatting system of the document caused this issue, which persisted despite correction efforts by MDPI Author Services.

Comments 6: 6) Item 3.6, line 11:  “Eleven SRs [3,10,24,27,28,30,31,33,34,38,39] included reported that there were an as”   Please, revise the sentence

Response 6: The sentence has been revised with the support of MDPI Author Services and now reads: “Eleven of the included SRs [3,10,24,27,28,30,31,33,34,38,39] reported an association between PD and DM, while two SRs [32,36] reported no association.

Comments 7: 7) Item 3.8: “Two SR [25,29] included reported that there is an association between PD and DM. This study meta-analyzed its results, where they found that the standardized mean difference ranged from 0.46 (CI: 0.08 to 0.84) [29] to 0.63 (CI: 0.39 to 0.87) [25]”. Please revise the sentence

Response 7: The sentence has been revised with the support of MDPI Author Services and now reads: “Two included SRs [25,29] that conducted meta-analyses reported an association between PD and DM, with a standardized mean difference ranging from 0.46 (CI: 0.08 to 0.84) [29] to 0.63 (CI: 0.39 to 0.87) [25].

Comments 8: 8) Discussion, line 87:  “they did not use a search strategy was exhaustive”… Please check this sentence

Response 8: The sentence has been revised with the support of MDPI Author Services and now reads: “did not use an exhaustive search strategy.

Comments 9: 9) Item 4.1, lines 122-125: Furthermore, it should be noted that of the included studies: four [10,27–29] analyze  T1DM, with only one [10] having high overall confidence; three [24,36,38] analyzing GDM  where all have high overall confidence; and ten [3,25,26,30–33,35,37,39] analyzing DM in general, eight [3,25,26,30,33,35,37,39] of which They have high general confidence.” Please review this sentence

Response 9: The sentence has been revised with the support of MDPI Author Services and now reads: “Furthermore, the following should be noted regarding the included studies: four [10,27–29] analyzed T1DM, with only one [10] demonstrating high overall confidence; three [24,36,38] analyzed GDM, with all demonstrating high overall confidence; and ten [3,25,26,30–33,35,37,39] analyzed DM in general, with eight [3,25,26,30,33,35,37,39] classified as having high general confidence.

Reviewer 3 Report

Comments and Suggestions for Authors

The current study is well-organised, exciting, and justified. It is indeed suitable for publication.

Inaccurate noticed:

No results in the abstract.

No null hypothesis has been formulated.

There are no references on lines 69-73, 74-78, 79-84, 84-92.

Author Response

 Reviewer 3

Comments 1: The current study is well-organised, exciting, and justified. It is indeed suitable for publication.

Inaccurate noticed:

No results in the abstract.

Response 1: Thank you very much for your positive feedback on our study. We appreciate your observation regarding the abstract. We have revised it and included the results, following the corrections made by MDPI Author Services. The updated abstract is as follows:

"Aim: To determine the clinical association between periodontal disease and diabetes mellitus through an umbrella review. Materials and Methods: A search for publications up to August 2023 was conducted using the following electronic databases: PubMed, Cochrane Database, Scopus, SciELO, Google Scholar, and OpenGrey. We included systematic reviews (SRs) with or without meta-analysis evaluating primary studies that investigated the association between periodontal disease and diabetes mellitus, and there were no time or language restrictions. Literature or narrative reviews, rapid reviews, intervention studies, observational studies, preclinical and basic research, abstracts, comments, case reports, protocols, personal opinions, letters, and posters were excluded. The AMSTAR-2 tool was used to determine the methodological quality of the included studies. Results: The preliminary search yielded a total of 577 articles, of which only 17 remained after discarding those that did not meet the selection criteria. Following their analysis, an association between periodontal disease and diabetes mellitus (type 1 and type 2 diabetes mellitus and gestational diabetes mellitus) was found. Conclusion: The findings and conclusions of this umbrella review indicate with high confidence that periodontal disease is associated with the onset of type 1 and type 2 diabetes mellitus and gestational diabetes."

Comments 2: No null hypothesis has been formulated.

Response 2: Thank you for raising this point. Since this study is an umbrella review, the primary aim is to synthesize existing evidence rather than test a specific hypothesis. However, if the inclusion of a null hypothesis is considered beneficial for clarity, we could introduce one to explicitly state that this study examines whether a consistent association exists between periodontal disease and diabetes mellitus. We are open to any guidance on the optimal way to incorporate this.

Comments 3: There are no references on lines 69-73, 74-78, 79-84, 84-92.

Response 3: We appreciate your careful attention to detail. In these sections, the statements are part of the discussion and provide an overall analysis based on findings from the studies included in the review. They are intended to give an overview of the general trends and limitations observed. Here is a summary of these statements:

An exhaustive literature search was conducted in the present study to summarize the available SRs investigating the association between PD and DM (T1DM, T2DM, and GDM), identifying 18 SRs that matched the inclusion criteria and were included in the analysis. SRs represent the highest level of the scientific evidence pyramid, but their results should be evaluated cautiously due to the potential for bias.

The SRs included in this study exhibit certain limitations related to the selected primary studies, such as the inclusion of different study types, the selection criteria for inclusion of studies, the inclusion of different population groups (children, adolescents, and adults), different diagnostic criteria for periodontal diseases, and the evaluation of different types of diabetes mellitus (T1DM, T2DM, and GDM).

On the other hand, more than 50% of the included studies demonstrated a high level of confidence, potentially enhancing the level of evidence for the results and conclusions presented in this study. However, the continued publication of SRs lacking a high level of confidence highlights the need for greater rigor in the development of such studies on this topic.

The AMSTAR-2 instrument, in its most current version, was used to assess the methodological quality of the included SRs. A factor related to the methodological quality that deserves emphasis is the critical domains 2, 4, 7, 9, 13, and 15 of AMSTAR-2. Some SRs failed to explicitly assess their methods, did not use an exhaustive search strategy, did not provide justification for the exclusion of studies, did not use a satisfactory technique to assess risk of bias, and did not consider the risk of bias of the included studies when interpreting or discussing their results. In addition, they did not report the risk of publication bias. This highlights a need for these elements to be included in the development of future SRs.

As this discussion addresses general insights and limitations found in prior reviews, we chose not to add specific references to maintain a cohesive narrative. However, we are open to adding citations if deemed necessary.

Reviewer 4 Report

Comments and Suggestions for Authors

Reviewer comments healthcare-3273480

General:

In this study, the authors have basically used a systematic approach to compile relevant data on the association of periodontal disease and diabetes and presented it in an organized manner, comparing and analysing the various parameters in a narrative fashion, as reported in the different systematic reviews.

Abstract:

-        Line 19- Clinical performance association of what? Please clarify? It is very vague currently.

-        Lines31-32- The conclusions do seem tom match the set objectives. Please relate clearly.

Introduction/ Background:

-         The introduction covers relevant literature and background related to the topic.

Materials & Methods:

- The study methodology is organised and well executed. It is is simple and clear.  

Discussion:

- There are many other confounding factors which influence the association of periodontal disease and diabetes mellitus. These elements should be discussed and limitations highlighted.        

Conclusions:

The conclusions need to balanced, and need to state the number of systematic review studies which support and deny the association. Currently, the extrapolation seems a bit unwarranted.    

Author Response

Reviewer 4

Comments 1:

General:

In this study, the authors have basically used a systematic approach to compile relevant data on the association of periodontal disease and diabetes and presented it in an organized manner, comparing and analysing the various parameters in a narrative fashion, as reported in the different systematic reviews.

Abstract:

-        Line 19- Clinical performance association of what? Please clarify? It is very vague currently.

-        Lines31-32- The conclusions do seem tom match the set objectives. Please relate clearly.

Response 1: Thank you for your comments and suggestions. We have revised line 19 with the support of MDPI Author Services, and it now reads as follows:
"Aim: To determine the clinical association between periodontal disease and diabetes mellitus through an umbrella review."

Regarding lines 31-32 in the conclusion, we have also made the necessary adjustments. It now reads:
"Conclusion: The findings and conclusions of this umbrella review indicate with high confidence that periodontal disease is associated with the onset of type 1 and type 2 diabetes mellitus and gestational diabetes."

Comments 2:

Introduction/ Background:

The introduction covers relevant literature and background related to the topic.

Materials & Methods:

- The study methodology is organised and well executed. It is is simple and clear.  

Discussion:

- There are many other confounding factors which influence the association of periodontal disease and diabetes mellitus. These elements should be discussed and limitations highlighted.

Response 2: Thank you for this observation. We have addressed this concern by emphasizing limitations related to the studies included in our review. Specifically, we have noted that certain primary studies varied in aspects such as the types of populations analyzed (e.g., children, adolescents, adults), differences in diagnostic criteria for periodontal disease, and differences in the types of diabetes mellitus evaluated (type 1, type 2, and gestational diabetes). These variations introduce potential confounding factors that could influence the association between periodontal disease and diabetes mellitus.

In addition, we discuss the limitations of certain SRs regarding methodological rigor, particularly in areas highlighted by the AMSTAR-2 assessment (e.g., failure to employ exhaustive search strategies, lack of clear justification for study exclusions, and inadequate risk-of-bias assessments). These limitations may affect the reliability of the conclusions drawn in some of the included studies, underscoring the importance of further research that addresses these confounders more comprehensively.

Comments 3:

Conclusions:

The conclusions need to balanced, and need to state the number of systematic review studies which support and deny the association. Currently, the extrapolation seems a bit unwarranted.

Response 3: We have revised the conclusion in line with the feedback, ensuring a more balanced and comprehensive summary of the evidence regarding the association between periodontal disease (PD) and diabetes mellitus (DM). The modified conclusion, based on the findings of our umbrella review and supported by MDPI Author Services, now reads as follows: “According to the findings of this umbrella review, there is strong evidence supporting the association between periodontal disease and various forms of diabetes mellitus, including type 1 and type 2, and gestational diabetes. The reviewed studies demonstrate a consistent relationship across different clinical parameters, such as plaque index, gingival index, clinical attachment level, probing depth, and bleeding on probing, with diabetes patients showing a higher likelihood of experiencing worsened periodontal conditions. However, two studies did not find an association, highlighting the need for further research to address potential differences in study populations and methodologies.”
